# Screening of Some Romanian Raw Honeys and Their Probiotic Potential Evaluation

Claudia Pașca [1,*], Liviu Alexandru Mărghitaș [1,†], Ioana Adriana Matei [2], Victorița Bonta [1], Rodica Mărgăoan [3], Florina Copaciu [4], Otilia Bobiș [1], Maria Graça Campos [5,6] and Daniel Severus Dezmirean [1,*]

1 Department of Apiculture and Sericulture, Faculty of Animal Sciences and Biotechnologies, University of Agricultural Sciences and Veterinary Medicine Cluj-Napoca, Calea Mănăștur 3-5, 400372 Cluj-Napoca, Romania; lmarghitas@usamvcluj.ro (L.A.M.); victorita.bonta@usamvcluj.ro (V.B.); obobis@usamvcluj.ro (O.B.)

2 Department of Microbiology, Faculty of Veterinary Medicine, University of Agricultural Sciences and Veterinary Medicine Cluj-Napoca, Calea Mănăștur 3-5, 400372 Cluj-Napoca, Romania; ioana.matei@usamvcluj.ro

3 Advanced Horticultural Research Institute of Transylvania, Faculty of Horticulture, University of Agricultural Sciences and Veterinary Medicine Cluj-Napoca, 400372 Cluj-Napoca, Romania; rodica.margaoan@usamvcluj.ro

4 Department of Chemistry and Biochemistry, Faculty of Animal Sciences and Biotechnologies, University of Agricultural Sciences and Veterinary Medicine, 400372 Cluj-Napoca, Romania; florina.copaciu@usamvcluj.ro

5 CQ-Centre of Chemistry—Coimbra, Department of Chemistry, Faculty of Sciences and Technology, University of Coimbra, Rua Larga, 3004-535 Coimbra, Portugal; mgcampos@ff.uc.pt

6 Laboratory of Pharmacognosy, Faculty of Pharmacy, Health Sciences Campus, University of Coimbra, Azinhaga de Santa Comba, 3000-548 Coimbra, Portugal

* Correspondence: claudia.pasca@usamvcluj.ro (C.P.); ddezmirean@usamvcluj.ro (D.S.D.)

† This co-author died on 14 May 2021.

**Abstract:** This study aimed to characterize raw honeys from different geographical origins in Romania, in respect of chemical composition, microbiological examination and evaluate their probiotic potential. The physico-chemical determinations were performed in APHIS-DIA Laboratory, Cluj-Napoca, Romania, using standard validated methods. Bacterial identification was performed for each sample and each colony type using Vitek® 2 Compact 15 system and PCR amplification using 16S rDNA bacterial universal primers (27F, 1492R), species being confirm by sequences analysis. In five raw honey samples, we have identified probiotic bacteria, such as: *Bacillus mycoides*, *Bacillus thuringiensis*, *Bacillus amyloliquefaciens*, *Bacillus subtilis*, and *Bacillus velezensis*. Generally, all honey samples meet the standard values for chemical composition. However, one sample having 7.44% sucrose was found to have also probiotics bacteria from the genus Bacillus because sucrose is a substrate for probiotics development. In conclusion, the Romanian raw honey can be a potential reservoir of probiotics, which confer a health benefit for consumers.

**Keywords:** honeycombs; fructooligosaccharides from honey; probiotic *Bacillus* strains; physicochemical characterization; antimicrobial activity

## 1. Introduction

Romania has an ancient tradition of beekeeping, and it is one of the most important honey producers in Europe, due to the variety of landforms, as well as the diverse and very rich flora. In the Romanian flora, there is a series of meliferous plants that stand out through a high honey production [1]. Honey, which is the earliest sweetener known to mankind, was used also for wound healing and as a traditional medicine before the advent of modern antibiotics, and now it is regarded not only as a natural sweetener but also as a healthy food with medicinal properties [2–4], and it has evoked a renewed interest with the reported upsurge in antibiotic resistance globally [5]. According to the definition set by European Union Council Directive 2001/110/EC [6], "Honey is the natural sweet substance produced by the *Apis mellifera* bees from the nectar of plants or from the secretions of living

parts of plants or excretions of plant sucking insects on the living parts of plants which the bees collect, transform by combining with specific substances of their own, deposit, dehydrate, store and leave in honeycombs to ripen and mature." Its composition is rather variable and primarily depends on the botanical and geographical origin of the floral source, although certain external factors also play an important role, such as seasonal and environmental factors and its processing.

Honey is a natural prebiotic composed of sugars, amino acids, enzymes, vitamins, and minerals [7]. Monosaccharides, glucose and fructose, are the major sugars (nearly 75%) and disaccharides, sucrose, maltose, turanose, isomaltose, and maltulose, are in the small amount in honey [8]. Non-digestible carbohydrates (oligosaccharides and polysaccharides), some peptides and proteins, and certain lipids (esters and ethers) are considered prebiotics [9]. Prebiotics are non-digestible food ingredients that beneficially affect the host by selectively stimulating the growth and/or activity of one or a limited number of probiotic bacteria [10–14].

Fructooligosaccharides (FOS) are non-digestible carbohydrates, composed mainly of chains of fructose units with a terminal glucose molecule, such as fructans, oligofructans, glucofructans, inulins, or oligosaccharides, such as sucrose, inulobiose, and levanobiose [9,15]. Sucrose is a fructooligosaccharide, which is prebiotic, and prebiotics are substrates for develop probiotics. If the concentration in fructooligosaccharides is higher, the probiotics are developing fast. High sucrose values in honeys are related to its botanical origin, for example, the honey dew or different honey maturity stages, or high nectar flux or artificial feeding of honey bees with sugar syrup [16]. Honeybees' relationship with probiotic microorganisms starts from the early metamorphosis stage of the larvae (day 4), when the feed is changed to honey, replacing royal jelly, until day 21 when adult larvae is ready to eclosionate. In this process, the larvae are immunologically stimulated due to probiotics contained in honey [2,17]. For a long time, researchers believed that the source of lactic acid bacteria in honey was pollen and secretions of flowers that arrived in honey transported by honey bees. However, later studies proved that the lactic acid bacteria are present in the stomach of the honeybees; therefore, the source of lactic acid bacteria is the bee itself [7]. In the honey production process, the enzyme glucose oxidase is responsible for the glucose transformation in galacturonic acid. This causes the natural acidification of honey and, therefore, its preservation. There, the majority of pathogenic and spoilage microorganisms are inhibited [18]. Due to honey acidity, yeasts and lactic acid bacteria are the predominant microorganisms. Among the lactic acid bacteria, there are probiotic microorganisms, from *Bacillus* and many other genera, such as *Lactobacillus*, *Lactococcus*, *Bifidobacterium*, *Leuconostoc*, and *Pediococcus* [19,20]. In 2014, the International Scientific Association for Probiotics and Prebiotics (ISAPP) consensus statement ratified an earlier definition of probiotics outlined by Food and Agriculture Organization and World Health Organization (FAO/WHO, 2001), as 'live microorganisms that, when administered in adequate amounts, confer a health benefit on the host'. Therefore, honey can be used in honeybee feeding and also in human consumption. Probiotics maintain microbial balance in intestine, prevent and inhibit the growth of pathogenic bacteria, and cure various intestinal diseases [21].

In this context, in various reports, several probiotic bacteria were isolated from honey or honeybee, and then characterized. Wang et al. (2015) [22] reported *Bacillus amyloliquefaciens* and *Bacillus subtilis* as the dominant bacteria in honey bee gut [23] or honey as an antagonistic agent against *Paenibacillus larvae* and *Ascosphaera apis* [24]. In South East Asia, different probiotic products containing *Bacillus* stains, either as single or mixed with other *Lactobacillus* strains, are used as an alternative to conventional antibiotics. Therefore, there is a risk of transferring antibiotic resistance genes to pathogens in the gut of humans and animals and release of drug resistance genes to the environment through feces [25]. Whereas the probiotics do not carry the risk for antibiotics resistance, they are used exactly for the opposite: the organism shall not develop resitance towards antibiotics.

The aim of the present study was to characterize the raw honeys from different geographic area and evaluate the probiotic potential of bacteria isolated from these honeys.

## 2. Materials and Methods

### 2.1. Honey Samples

Raw honey was harvested directly from honeycombs that were collected from different Romanian apiaries in autumn 2020 (Figure 1): five samples (P1- Sălaj area; P2-Râșca apiary; P3-Țaga apiary; P5-Coruşu apiary; and P6-USAMV apiary) from north-west of the country, two samples (P7-Harghita area and P10-Alba area) from the center of the country, one sample (P4-Maramureș area) from north of the country, another sample (P9-Banat area) from west of the country, and the last one sample (P8-Tulcea area) from south-east of the country. In this research, we choose to study honey because it is the most consumed bee product. The characteristics of these honey samples (P1–P10) were analyzed in accordance with the FDA (Food and Drug Administration) standard methods or scientific articles. All the chemical reagents and microbial media used in this research were purchased from Merck (Darmstadt, Germany) and Sigma-Aldrich (Saint Louis, MO, USA) companies, with analytical grade.

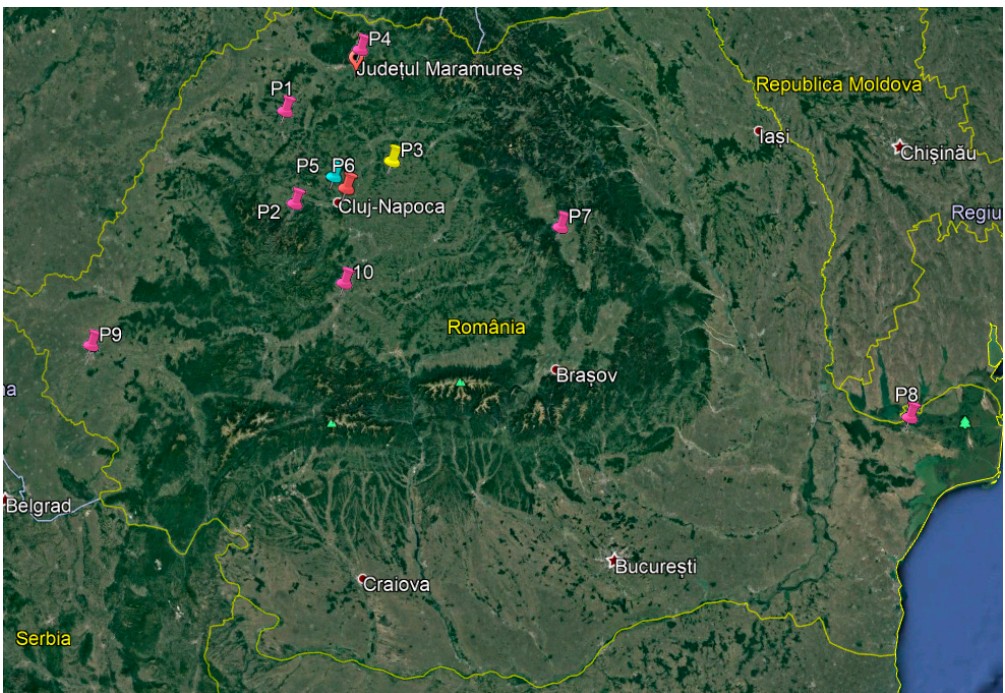

**Figure 1.** Geographical spread of raw honey samples (P1–P10).

### 2.2. Physico-Chemical Characterization of Raw Honey Samples

#### 2.2.1. Nutritional Parameters of Raw Honey Samples

The *melissopalynological analysis* was made after the method of Louveaux et al. (1978) [26], without acetolysis. Briefly, 10 g of honey was dissolved in diluted $H_2SO_4$ (5‰), centrifugated 10 min (3396 rcf), and the sediment washed twice with distilled water. After another centrifugation, the supernatant was discharged, and two drops of liquid, together with the entire sediment, was put on a microscopic slide, and an area $20 \times 20$ m$^2$ was mounted with glycerine-gelatine-fuxine. Slide examination was performed using an Olympus BX51microscope at 1000×magnification (for identification) and 400×magnification for counting. One thousand pollen grains were counted from every slide, and percentages of different botanical species were calculated. For the identification of pollen type, different pollen guides were used.

*Moisture content:* The method used for moisture content was adapted from International Honey Commission (IHC) [27], measuring the refractive index of a liquid honey sample, using Abbe refractometer (Abbé WAY-S Selecta—Spain). In the case of a crystalized sample, honey was placed in a tube with rod and was heated in a water bath at 40 °C until liquification. The refractive index read at 20 °C on the refractometer is transformed in water content using the table from the standard. Results are expressed as % water.

*Sugars profile determination:* HPLC (High-performance liquid chromatography) analysis of the carbohydrates is carried out on a modified Alltima Amino 100 Å stainless steel column (Alltech, Nicholasville, KY, USA) (4.6 mm diameter, 250 mm length, particle size 5 μm) following IHC method modified in APHIS-DIA Laboratory [28]. The SHIMADZU instrument (LC–10AD VP model, Shimadzu, Kyoto, Japan) was equipped with degasser, two pumps, auto sampler, thermostat oven, controller, and refractive index detector. The injection volume was 10 μL, and the flow rate was 1.3 mL/min. The mobile phase was acetonitrile/water (75:25 *v/v*). Briefly, 5 g of honey were dissolved in ultrapure water (40 mL), transferred into a 50 mL volumetric flask, containing 25 mL HPLC grade purity methanol, and subsequently filled to the mark with water. The honey solution was filtered through a 0.45 μm membrane filter, collected in HPLC glass vials for analysis. A calibration curve for each sugar was made using standard solutions of different concentrations (0.5–80 mg/mL) for the quantification of sugars, with regression coefficients $r^2$ higher than 0.998. Quantification was made by comparing the obtained peak area with those of standard sugars. The results were expressed as g/100 g honey.

*Total lipids*: The content of total lipids in honey samples was determined using Soxhlet method (Soxtherm, Gerhardt, Germany), a literature method [29] modified in APHIS-DIA Laboratory. Briefly, 2 g of honey samples were weighted on filter paper, and also the extraction glasses containing 2 boiling stones, will be weighted and, after that, together with the cartridge and the solvent (90 mL etil-eteric), will be fixed in PTFE cylinders. The method is set from the multistat: extraction temperature 140 °C, extraction time 5 h, washing 30 min, solvent evaporation in hot air flow 10 min. After extraction ends, extraction glasses will be placed in the oven at 60 °C, for one half hour, to eliminate traces of solvent and, after cooling, will be weighted, and the results were expressed as percent.

*Total proteins:* The content of total proteins of honey samples was determined using Bradford method. The original method of Bradford [30] was modified in our laboratory, for honey analysis. Briefly, 2 g of honey samples dissolve with 2 mL of ultrapure water in a 10 mL Berzelius glass. Take 0.1 mL of the diluted honey solution and place in a test tube, over which 2 mL Bradford reagent is added. Thus, the sample obtained shall be homogenized. The absorption of solutions is measured at 595 nm. The amount of protein was calculated according to the calibration curve.

### 2.2.2. pH and Acidity Value Measurement

Honey has an acidic pH due to the presence of different organic acids in the composition. This low pH inhibits the presence and growth of different unwanted microorganisms, and the presence of different acids help developing the flavor and aroma or different types of honey. Honey pH is measured with an automatic titrator (Titroline Easy SCHOTT Germany) from a solution of 10% (*w/v*) honey. Free acidity of honey is obtained from neutralization with a solution of sodium hydroxyde. Lactonic acidity is obtained by titration with NaOH in excess and making the neutralization curve of the NaOH excess by titration with sulfuric acid. Total acidity of honey is the sum of free acidity and lactonic acidity. The method from International Honey Commission [27] and adapted for the automatic titrator in our laboratory. Acidity is expressed as meqNaOH/kg honey.

### 2.2.3. Mineral Content

To determine the levels of micro and macro elements: Ni, Na, Cd, Mg, K, Cr, Ca, Fe, Mn, Cu, Se, Pb, and Zn from studied honey samples, the atomic absorption spectrometry method was used [31,32]. The mineralization of the samples was made in a microwave

furnace, Berghof digestion system MWS-2. Approximately 0.3 g of the homogenized honey samples were placed in special Teflon tubes, 2 mL of 65% $HNO_3$, after which 3 mL of $H_2O_2$ was added before the container was sealed. At the end of the initiated program, the solution is transferred into graded plastic containers and the sample is diluted with ultrapure water to a volume of 30 mL. An AAnalyst 800 Atomic Absorption Spectrometer from Perkin-Elmer (Shelton, CT, USA) equipped with a cross-linked graphite furnace was used. The absorption wavelength for the determination of each micro and macro element, together with its linear working range and correlation coefficient of calibration graphs, were calculated. The reagents used to perform the analyses were of analytical purity, namely 65% nitric acid and ultra-purified water. For the calibration curve, following dilutions of standard solutions were used: chromium (10 µg/L), manganese (10 µg/L), calcium (2 µg/L), nickel (50 µg/L), iron (20 µg/L), cadmium (2.0 µg/L), sodium (4.0 µg/L), potassium (5.0 µg/L), zinc (2.0 µg/L), selenium (100 µg/L), magnesium (1.0 µg/L), copper (25 µg/L), and lead (50 µg/L) [32]. The result obtained represents the concentration of mineral elements expressed in µg/L, then transformed according to the amount of weighted sample.

### 2.2.4. Quality Analysis of Raw Honey Samples

*Hydroxymethylfurfural content:* The concentration of hydroxymethylfurfural (5-hydroxy methil-furan-2-carbaldehide—HMF) content was determined via high performance liquid chromatography with photodiode array detection (HPLC-PDA) [27] (Shimadzu VP system—Japan). Operational parameters of chromatographic system: mobile phase ultrapure water (solvent A), methanol (solvent B), gradient scheme: 10% B at 0–13 min, 90% B at 13–15 min, 90% B at 20 min, and 90–10% B at 21 min, equilibrating the column (10% B) until 30 min, flow rate 0.8 mL/min, injection volume 20 µL, Discovery HS C18, 5 µm 4.6 × 250 mm column, column temperature 30 °C, detection at 283 nm. Ten grams of honey is weighted into a Berzelius glass and dissolved in approximately 25 mL of distilled water. Honey solution is transferred quantitatively into a 50 mL volumetric flask. Add 0.5 mL of Carrez I solution (15 g of potassium hexacyanoferrate (II), $K_4Fe(CN)_6 \times 3H_2O$ in water and make up to the mark in 100 mL) and mix. Then, add 0.5 mL of Carrez II solution (30 g of zinc acetate, $Zn(CH_3COO)_2 \times 2H_2O$, and make up to 100 mL), mix, and make up to 100 mL with water. Filter through filter paper, removing the first 10 mL. The filtrate was afterwards filtered into glass vial through a 0.45 µm syringe filter and inject into HPLC system. Different concentrations of HMF (Hidroximetilfurfural) standard (1–50 mg/mL) were injected into the chromatographic system and a calibration curve was constructed. The concentration of HMF from the samples is calculated with the equation:

$$HMF(mg/kg) = \frac{Cc \times 5 \times 10}{m},$$

where:

Cc—concentration from the calibration curve,
*m*—mass of weighted honey, and
results are expressed in mg/kg honey.

*Diastase activity:* For the determination of diastazic activity, Amylazime—Megazyme spectrophotometric method was used [33]. Amylazime tablets contain amylose (the linear fraction of starch). In the presence of α-amylase (from the honey solution), the substrate is hydrolyzed, and the soluble colored product is released. The reaction is terminated by the addition of Trizma solution, followed by filtration. The absorbance of the filtrate is read at 590 nm towards a control sample (without honey) as blank. There is a linear correlation between absorbance of the filtrate and the diastase activity in the sample. For sample preparation, 2 g of honey was weighted and dissolved in 40 mL of 100 mM maleate buffer (pH 5.6) in a 50 mL Berzelius glass. The solution was then transferred quantitatively into a 50 mL volumetric flask and make up to the mark with water. One mL of the diluted honey solution was placed in a test tube and pre-incubated at 40 °C for 5 min. One tablet

of Amylazime was added to the tube without removing the test tube from the water bath. The tablet hydrated quickly and absorbed most of the liquid.

The samples were then incubated the tube at 40 °C for exactly 10 min; afterwards, 10 mL of basic Trizma solution (2% *w/v*) was added to complete the reaction, and the test tube was shacked vigorously and stored at room temperature for 5 min. The content of the tubes was filtered through paper filter and read at 590 nm on a spectrophotometer (UV-1700 Shimadzu Instruments) against a control.

Diastazic activity was calculated by the following formula: $20 \times DO_{590}$, where $DO_{590}$ represents the absorbance. Results are expressed in Schade units/g honey.

*Contaminants: tetracycline and oxytetracycline:* Tetracycline and oxytetracycline from honey were determined using high performance liquid chromatography with photodiode array detection (HPLC-PDA). The method was adapted from AOAC methods [34] in our laboratory for the detection and quantification of antibiotics from honey. Sample preparation was based on the purification and extraction of tetracycline residues from honey using affinity columns with metallic chelates [35]. The chromatographic system (Shimadzu VP Series, Japan) consists of: binary pump, degasser, system controller, column oven, auto-injector, and detector set at 360 nm. The separation was performed on a Nucleosil 100 RP-18, 5 μm column, 250 × 4.6 mm ID, equipped with guard column. The mobile phase is 10 mM oxalic acid, acetonitrile, and methanol (16/3/2). The flow rate was 1 mL/min, injection volume: 50 μL and column temperature: 35 °C. For sample preparation, an Alltech vacuum Manifold with polypropylene mini-columns of 3 mL with 10 mL reservoir, containing a frit at the bottom (Supelco), was used. The extraction solution is sodium succinate buffer 0.1 M, eluent McIlvine-EDTA-NaCl buffer, and chelating Sepharose fast flow resin in 20% methanol suspension [35]. From a stock solution of 0.1 mg/mL oxytetracycline and tetracycline, different dilutions were made (5–75 μg/L) and were injected separately in duplicate. Following the chromatograms registrations, calibration curves were constructed, to be used in unknown concentration determination from the honey samples. The oxytetracycline or tetracycline residue content of the sample was calculated by comparing the signal area corresponding to the sample with that of the standard solution, taking into consideration the dilution. There is a direct proportionality between the concentration and the signal area given by those. Calculation was made following the equation:

$$\text{Residue concentration (μg/kg)} = (V_{elution}/m_{sample}) \times C_c,$$

where:

$V_{elution}$—elution volume from sample preparation (3 mL),
$m_{sample}$—honey sample mass (g), and
$C_c$—concentration from the calibration curve.

Every batch of analysis contained one pure standard mixture run and fortified samples to exclude uncertainty from inconclusive results. The results are expressed in μg/kg honey (ppm).

### 2.3. Microbiological Examination of Raw Honey Samples

From each lot (P1–P10), 25 g of honeycombs were weight and added to 225 mL Tween 80 tryptic soy broth under sterile condition. The samples were incubated 24 h at 37 °C in a shaker incubator. The next day, up to 6 10-fold serial dilution were performed, and 1 mL sample were poured on Tween 80 tryptic soy agar. For each sample dilution, 2 plates were inoculated. One of each dilution plates was cultivated in aerobic condition at 37 °C for 24 h, and the second one in anaerobic condition at 37 °C for 5 days [23].

The number of total germs (NTG) was established by counting at the proper dilution on the plates cultivated aerobically, and the bacterial concentration was calculated taking into account the dilution.

Bacterial identification was performed for each sample and each colony type using Vitek® 2 Compact 15 system (bioMérieux, Marcy l'Etoile, France). The appropriate card type (BCL, ANC, GP, and GN) was used accordingly to the microscopical identification (Gram staining) and manufacturer recommendations. Where Vitek® identification failed, the colony was processed further by DNA extraction using commercial kits (ISOLATE II Genomic DNA Kit, Bioline, UK) and PCR amplification using 16 s rDNA bacterial universal primers (27F, 1492R) using previously published protocols [36,37]. The amplification was carried out in 25 µL reaction mixture containing 12.5 µL of Green PCR Master Mix (Rovalab GmBH), 6.5 µL PCR water, 1 µL of each primer (0.01 mM), and 4 µL aliquot of isolated DNA. The PCR was carried out using a T100™ Thermal Cycler (Bio-Rad). The PCR products were visualized by electrophoresis in a 1.5% agarose gel stained with SYBR® Safe DNA gel stain (Invitrogen). Positive PCR products were purified using commercial kits (Isolate II PCR and Gel Kit, Bioline, UK). Sequencing analysis was performed (Macrogen Europe, Amsterdam), and the obtained sequences were edited and analyzed using Geneious® (Biomatters LTD) 4.8.7 and compared with those available in the GenBank database by BLASTn analysis.

### 2.4. Antibacterial Activity of Bacillus spp. Isolates

The antimicrobial activity of Bacillus spp. isolates (B. subtilis, B. thuringiensis 1, B. mycoides, B. amyloliquefaciens, B. velezensis 1, B. velezensis 2, B. thuringiensis 2, B. velezensis 3) was tested against 3 Gram-positive and 3 Gram-negative references strains of bacteria (Enterococcus faecalis ATCC 29212, Staphylococcus aureus ATCC 6538P, Listeria monocytogenes ATCC 13932, Escherichia coli ATCC 10536, Salmonella enteritidis ATCC 13076, Pseudomonas aeruginosa ATCC 27853). The tryptic soy gar plates were inoculated with 1 mL of 0.5 McFarland bacterial suspension. After drying, 9 holes were sterile cut, and 50 µL of Bacillus spp. 2 McFarland bacterial suspension was added. The pates were incubated aerobically at 37 °C for 24 h. The next day, the inhibition radius was measured [38].

### 2.5. Statistical Analysis

All determinations were made in series of three independent repetitions, and the obtained results were expressed as mean ± standard deviation (SD). Significant differences between samples were analyzed with one-way ANOVA (Analysis of varianc) post hoc tests, and pairwise multiple comparisons were conducted using Tukey's test. Significant differences were reported based on $p < 0.05$. Statistical analyses were performed using the SPSS programme (version 22.0, Chicago, IL, USA). Dendrograms were generated using the Euclidean distance based on Ward's algorithm for clustering [39]. To reduce the pollen dataset to a lower number of dimensions, cluster analyses using the Paleontological Statistics (PAST) software (version 2.17, Oslo, Norway, 2012) was performed [40].

## 3. Results

### 3.1. Melisopalinological Analysis

Melisopalinological analysis of raw honey samples provides information about the plants from which bees collected the nectar. In Table 1 are represented the families, genera and species to which the analyzed honey samples belong. From all the analyzed honey samples, in only five, a predominant pollen species (>45%) was identified (Asteraceae family), and, in five samples, secondary pollen species (16–45%) were identified, belonging to Asteraceae, Fabaceae, Boraginaceae, Laminaceae families and other important minor pollen and minor pollen.

**Table 1.** Melissopalynological analysis of the studied honey samples.

| Sample | Predominant Pollen (>45%) | | Secondary Pollen (16–45%) | | Important Minor Pollen (3–15%) | | Minor Pollen (<3%) | |
|---|---|---|---|---|---|---|---|---|
| | **Family** | **Specie** | **Family** | **Specie** | **Family** | **Specie** | **Family** | **Specie** |
| **P1** | Asteraceae J | *Centaurea montana* type | | | Asteraceae S Brassicaceae Lamiaceae Asteraceae A Fabaceae | *Cirsium* type *Brassica* sp. *Matricaria* sp. | Asteraceae T Asteraceae H Geraniaceae Poaceae Apiaceae Rosaceae Cornaceae Plantaginaceae Caryophyllaceae | *Taraxacum* sp. *Helianthus* sp. *Zea mays* * *Rubus* sp. *Cornus* sp. *Plantago* sp. * |
| **P2** | | | *Brassicaceae* | *Brassica* sp. | Ericaceae Rosaceae Lamiaceae Scophulariaceae Fabaceae Fagaceae | *Prunus* sp. *Filipendula* sp. *Quecus* sp. * | Asteraceae J Asteraceae H Plantaginaceae Asteraceae T Asteraceae A Rosaceae Poaceae* Salicaceae | *Centaurea montana* type *Helianthus* sp. *Plantago* sp. * *Taraxacum* sp. *Matricaria* sp. *Rubus* sp. *Salix* sp. |
| **P3** | | | Asteraceae H Asteraceae T Fabaceae | *Bellis* sp. *Taraxacum* sp. | Asteraceae Apiaceae Asteraceae S | *Artemisia* sp. * *Ambrosia* sp. * *Cirsium* type | Asteraceae A Asteraceae J Tiliaceae Ranunculaceae Salicaceae Brassicaceae Rosaceae Plantaginaceae Asteraceae H Scrophulariaceae Poaceae | *Matricaria* sp. *Centaurea* sp. *Tilia* sp. *Salix* sp. *Rubus* sp. *Plantago* sp. * *Helianthus* sp. *Zea mays* * |

Table 1. *Cont.*

| Sample | Predominant Pollen (>45%) | | Secondary Pollen (16–45%) | | Important Minor Pollen (3–15%) | | Minor Pollen (<3%) | |
|---|---|---|---|---|---|---|---|---|
| | Family | Specie | Family | Specie | Family | Specie | Family | Specie |
| **P4** | Asteraceae T | *Taraxacum* sp. | | | Asteraceae | *Ambrosia* sp. * | Brassicaceae Fabaceae Asteraceae A Caryophyllaceae Rosaceae Asteraceae S Salicaceae Asteraceae J Ericaceae | *Matricaria* sp. *Filipendula* sp. * *Cirsium* type *Salix* sp. *Centaurea montana* type |
| **P5** | | | Asteraceae J Fabaceae | *Centaurea montana* type | Lamiaceae Asteraceae S Apiaceae Rosaceae Scrophulariaceae | *Cirsium* type | Tiliaceae Asteraceae A Asteraceae H Asteraceae T Boraginaceae Geraniaceae Loranthaceae Caryophyllaceae | *Tilia* sp. *Matricaria* sp. *Helianthus* sp. *Taraxacum* sp. *Symphytum* sp. *Loranthus europaeus* |
| **P6** | | | Asteraceae J Asteraceae T | *Centaurea sp. montana type* *Taraxacum* sp. | Asteraceae S Asteraceae A Asteraceae Amaranthaceae Asteraceae | *Cirsium* type *Matricaria* sp. *Artemisia* sp. * *Ambrosia* sp. * | Brassicaceae Fabaceae Apiaceae Lamiaceae Poaceae Scrophulariaceae Asteraceae H | *Brassica* sp. *Zea mays* * *Bellis type* |
| **P7** | | | Boraginaceae Asteraceae H | *Echium* sp. *Helianthus* sp. | Asteraceae H Plantaginaceae Fabaceae Brassicaceae Scrophulariaceae Lamiaceae Asteraceae A Rosaceae | *Bellis type* *Plantago* sp.* *Brassica* sp. *Matricaria* sp. | Asteraceae Geraniaceae Asteraceae T Apiaceae Poaceae Asteraceae Poaceae * Asteraceae S Polygonaceae | *Ambrosia* sp. * *Taraxacum* sp. *Zea mays* * *Artemisia* sp.* *Cirsium* type *Rumex* sp. * |

Table 1. *Cont.*

| Sample | Predominant Pollen (>45%) | | Secondary Pollen (16–45%) | | Important Minor Pollen (3–15%) | | Minor Pollen (<3%) | |
|---|---|---|---|---|---|---|---|---|
| | Family | Specie | Family | Specie | Family | Specie | Family | Specie |
| **P8** | | | Asteraceae J<br><br>Asteraceae H<br>Lamiaceae | *Centaurea montana* type<br>*Helianthus* sp. | Fabaceae<br>Geraniaceae<br>Asteraceae S<br>Rosaceae<br>Asteraceae A<br>Asteraceae T<br>Brassicaceae<br>Plantaginaceae | *Cirsium* type<br>Rubus sp.<br>*Matricaria* sp.<br>*Taraxacum* sp.<br>*Brassica* sp.<br>*Plantago* sp. * | Asteraceae C<br><br>Poaceae<br>Onagraceae | *Centaurea cyanus*<br>*Zea mays* *<br>*Epilobium* sp. |
| **P9** | Asteraceae H | *Helianthus* sp. | | | Boraginaceae<br>Fabaceae<br>Rosaceae | *Echium* sp.<br><br>*Rubus* sp. | Asteraceae H<br>Boraginaceae<br>Rosaceae<br>Amaranthaceae<br>Asteraceae S<br>Brassicaceae<br>Betulaceae | *Bellis type*<br>*Echium* sp.<br>*Filipendula* sp. *<br><br>*Cirsium* type<br>Brassica sp.<br>*Corylus* sp. * |
| **P10** | Asteraceae | *Xanthium* sp.* | | | Cyperaceae *<br>Asteraceae T<br>Plantaginaceae | *Taraxacum* sp.<br>*Plantago* sp. * | Amaranthaceae<br>Asteraceae<br>Asteraceae A<br>Apiaceae<br>Rosaceae<br>Asteraceae S<br>Lamiaceae<br>Poaceae*<br>Ranunculaceae<br>Asteraceae | *Artemisia* sp. *<br>*Matricaria* sp.<br><br>*Rubus* sp.<br>*Cirsium* type<br><br><br><br>*Ambrosia* sp. * |

*—non-nectariferous plants.

In the same time, hierarchical clustering was performed in order to differentiate the honey samples based on their botanical and geographical origin (*r* = 0.95). As shown in Figure 2, the dendrogram obtained from the Cluster analysis (Euclidean distance) reveals the formation of two main clusters, except samples P10 and P4, which are considered 'outliers', not being grouped with the others, which might be based on the predominant pollen in Asteraceae (*Xanthium* sp.) and only a few minor pollens for P10. The same was noticed for sample P4 which is predominant in Asteraceae T (*Taraxacum* sp.), followed by sample P9 predominant in Asteraceae H (*Helianthus* sp.). The following sub-cluster comprises samples P2, P3, and P7, which are mainly polyfloral. The last sub-cluster comprises samples P6 and P8, which are polyfloral, but with secondary pollen in Asteraceae J. The same was noticed for samples P1 and P5, which are predominant in Asteraceae J and secondary in Fabaceae. This approach shows that cluster analysis can discriminate the honey samples based on their botanical origin, even among polyfloral samples.

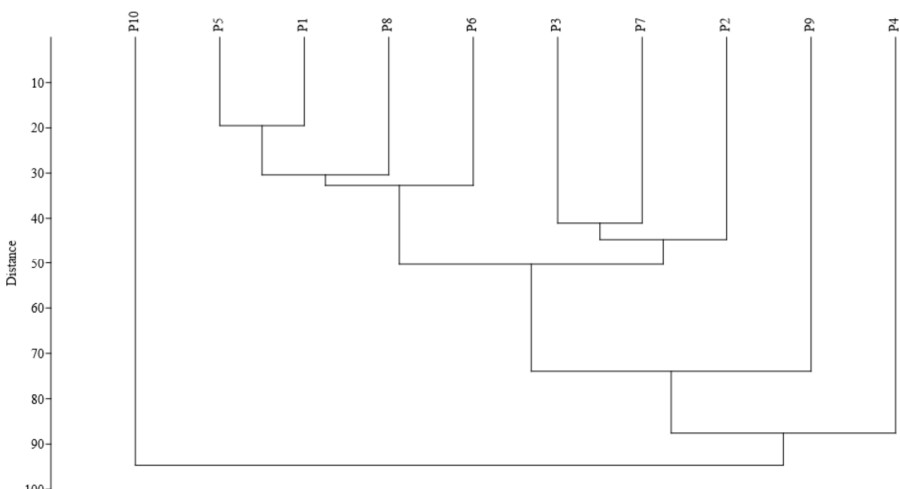

**Figure 2.** Hierarchical clustering of the honey samples based on melissopalynological analysis (Euclidean distance, *r* = 0.95).

## 3.2. Physico-Chemical Parameters of Raw Honey Samples

Nutritional parameters for raw honey samples were made using the methods described previously. Table 2 presents the identified free sugars, moisture, proteins, and lipids in the analyzed samples. Seven sugars (Figure 3) were identified in the raw honey samples, fructose and glucose being the predominant ones. Sucrose, turanose, maltose, trehalose, and erlose were detected in low amounts. The higher concentrations in fructose were identified in samples P8 and P9, while glucose was identified in higher amounts in sample P9. Besides the reducing sugar analysis, the amount of sucrose is a very important parameter in evaluating the honeys' maturity [7]. In addition, sucrose is an important fructooligosaccharide for the growing of the probiotic bacteria. Sucrose was quantified in higher amounts in sample P5 (7.44 ± 0.37%) (Figure 4), followed by sample P1 (3.06 ± 0.15%). The sugar and moisture results obtained in the present study are in accordance with those specified in national and European directive [41,42]. Only sucrose for P5 (7.44 ± 0.37%) exceeds more than 5 g/100 (probably the amounts in fresh honey was higher than in ripe honey). The protein content in analyzed honeys was in the range of 0.1–0.5% [16], and lipid content ranged between 0.07 and 0.42%.

**Table 2.** Sugar, moisture, proteins, and lipids results from raw honeys samples.

| Honey Samples | Fructose (%) | Glucose (%) | Sucrose (%) | Turanose (%) | Maltose (%) | Trehalose (%) | Erlose (%) | Moisture (%) | Proteins (%) | Lipides (%) |
|---|---|---|---|---|---|---|---|---|---|---|
| P1 | 38.8 (2.02) [a] | 32.8 (1.57) [cb] | 3.06 (0.15) [b] | 0.83 (0.04) [g] | 2.4 (0.11) [d] | 0.17 (0.01) [e] | 0.78 (0.04) [c] | 16.0 (0.78) [cb] | 0.12 (0.00) [d] | 0.07 (0.00) [g] |
| P2 | 38.1 (1.95) [a] | 32.2 (1.53) [cb] | 0.39 (0.01) [e] | 2.27 (0.11) [a] | 2.8 (0.14) [dc] | 1.06 (0.05) [a] | 0.95 (0.04) [c] | 16.4 (0.80) [cb] | 0.29 (0.01) [b] | 0.30 (0.01) [b] |
| P3 | 37.7 (1.91) [a] | 32.8 (1.56) [cb] | 1.93 (0.09) [c] | 1.64 (0.08) [c] | 3.94 (0.19) [a] | 0.39 (0.02) [d] | 2.06 (0.10) [b] | 14.4 (0.71) [c] | 0.17 (0.00) [c] | 0.21 (0.01) [d] |
| P4 | 36.5 (1.85) [a] | 33.5 (1.61) [cb] | 1.33 (0.06) [d] | 1.79 (0.08) [c] | 3.37 (0.16) [b] | 0.54 (0.02) [c] | 2.20 (0.10) [b] | 16.1 (0.76) [cb] | 0.06 (0.00) [e] | 0.19 (0.01) [ed] |
| P5 | 36.2 (1.82) [a] | 29.4 (1.41) [c] | 7.44 (0.37) [a] | 1.49 (0.07) [dc] | 3.25 (0.16) [b] | 0.39 (0.02) [d] | 3.13 (0.16) [a] | 14.8 (0.74) [cb] | 0.10 (0.00) [d] | 0.16 (0.01) [fe] |
| P6 | 38.5 (1.90) [a] | 32.1 (1.63) [cb] | 1.01 (0.04) [d] | 1.97 (0.09) [b] | 3.19 (0.15) [cb] | 0.80 (0.04) [b] | 2.28 (0.11) [b] | 17.0 (0.82) [ba] | 0.02 (0.00) [f] | 0.15 (0.00) [fe] |
| P7 | 37.6 (1.85) [a] | 34.1 (1.74) [cb] | 1.07 (0.05) [d] | 1.31 (0.06) [ed] | 3.32 (0.16) [b] | 0.47 (0.02) [dc] | 0.75 (0.04) [c] | 16.7 (0.79) [ba] | 0.21 (0.01) [c] | 0.29 (0.01) [b] |
| P8 | 40.1 (1.95) [a] | 34.1 (1.68) [b] | 0.42 (0.02) [e] | 1.10 (0.05) [fe] | 2.52 (0.13) [d] | 0.22 (0.01) [e] | 0.37 (0.02) [d] | 16.0 (0.82) [cb] | 0.28 (0.01) [b] | 0.25 (0.01) [c] |
| P9 | 40.1 (1.94) [a] | 39.4 (1.89) [a] | 0.08 (0.00) [e] | 0.27 (0.01) [h] | 1.14 (0.05) [e] | 0.40 (0.02) [d] | nd | 15.9 (0.82) [cb] | 0.18 (0.01) [c] | 0.42 (0.02) [a] |
| P10 | 36.9 (1.89) [a] | 31.9 (1.52) [cb] | 0.18 (0.01) [e] | 1.01 (0.05) [gf] | 3.87 (0.19) [a] | 0.25 (0.01) [e] | nd | 19.0 (0.95) [a] | 0.48 (0.02) [a] | 0.15 (0.01) [f] |

The values are mean of three samples (n = 3, standard deviation in parentheses), analyzed individually in triplicate. Different letters within a column denote significant differences ($p < 0.05$); nd = not detectable.

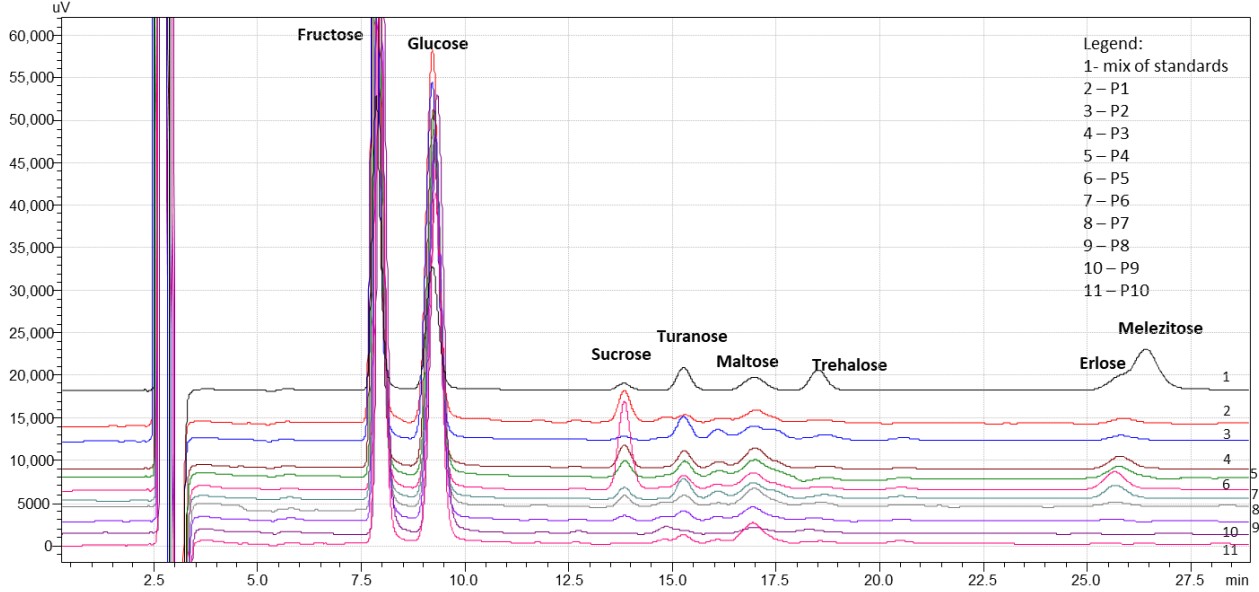

**Figure 3.** Chromatogram of the identified carbohydrates in the raw honey samples.

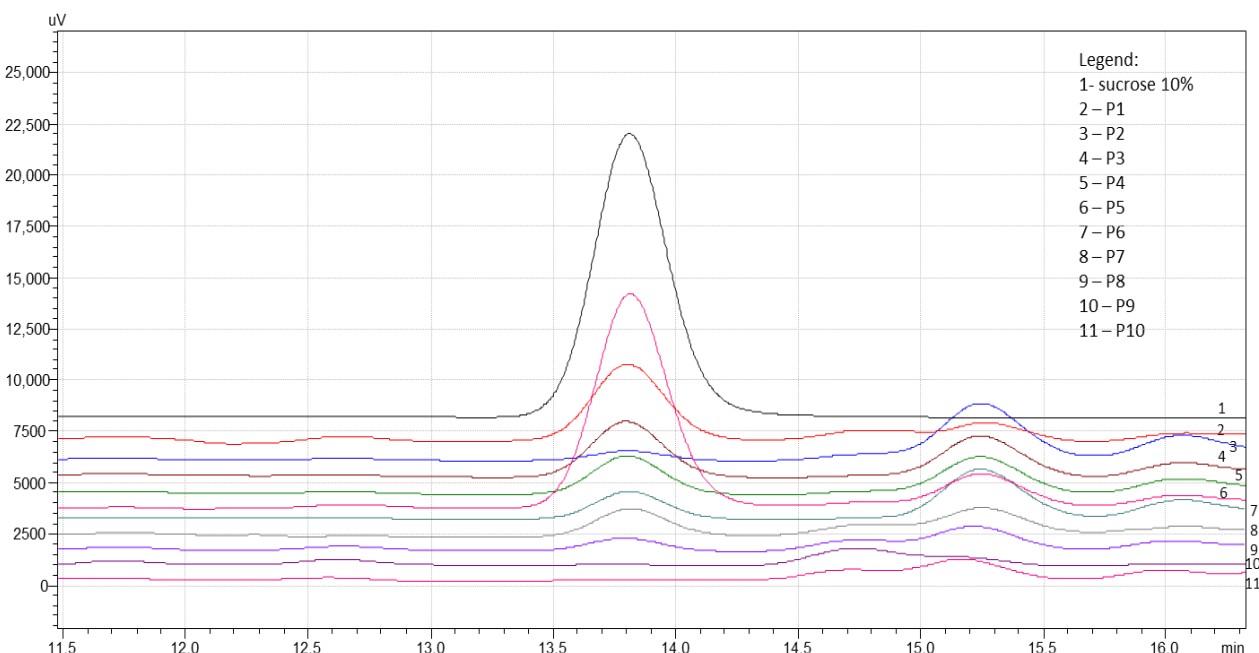

**Figure 4.** Chromatogram of sucrose standard solution 10% (black) and analyzed samples (colored signals).

A total of 13 micro and macro elements were identified and quantified in the analyzed samples (Table 3). Mineral content in honey is generally low, ranging between 0.02 and 0.3% in blossom honeys, while, in honeydew, honeys can reach 1% of the total [43,44]. Potassium is the main one, standing for 80% of the total, because of its quick secretion in nectar sources [45]. The amount of minerals present in honey does not significantly contribute to the dietary recommendations [16]. In the present study, higher amounts of potassium was identified (375.30 ± 19.3 mg/kg), as well as calcium (83.0 ± 4.26 mg/kg) and magnesium (69.7 ± 3.59 mg/kg), in P1 sample. Low concentrations of selenium (0.13–0.79 mg/kg), zinc (0.32–0.50 mg/kg), and manganese (0.07–7.26 mg/kg) were identified. Other microelements identified were Cu, Fe, Cr, and Ni, only in sample P3. The level of Na was below the detection limit of the method (4.0 μg/L). No heavy metals, such as cadmium and lead, were identified in the analyzed raw honey samples.

As shown in Figure 5, the dendrogram obtained from the Cluster analysis (Euclidean distance) reveals the formation of two clusters, with the exception of sample P1, which is an 'outlier', being distinct from the others due to its high content in the minerals K, Ca, and Mg. At the same time, a high content in sucrose and lower in lipids and trehalose is noticed. The following cluster comprises the samples P8 and P5, followed by samples P10 and P4, which display similar concentrations in fructose, glucose, and lipids, as well as in Ca, Se, and Cr. The next sub-cluster comprises samples P7, P9, and P3, which have similar protein values, as well as in the mineral K. The last sub-cluster comprises samples P6 and P2, with relatively close values in fructose, glucose, and moisture content. The same is noticed in the minerals Ca, K but also in those with lower values, such as Zn and Cr.

**Table 3.** Minerals results from raw honey samples.

| Honey Samples | Ca (mg/kg) | Mg (mg/kg) | Cu (mg/kg) | Fe (mg/kg) | Mn (mg/kg) | K (mg/kg) | Ni (mg/kg) | Se (mg/kg) | Zn (mg/kg) | Cr (mg/kg) |
|---|---|---|---|---|---|---|---|---|---|---|
| P1 | 83.0 (4.26) [a] | 69.7 (3.59) [a] | 0.31 (0.01) [b] | 7.96 (0.38) [b] | 2.23 (0.11) [b] | 375.30 (19.3) [a] | nd | 0.79 (0.04) [a] | 0.39 (0.02) [c] | 0.37 (0.02) [a] |
| P2 | 33.6 (1.73) [d] | nd | 0.21 (0.01) [dc] | 3.58 (0.18) [c] | 7.26 (0.38) [a] | 282.64 (14.6) [b] | nd | 0.40 (0.02) [b] | 0.33 (0.01) [d] | 0.20 (0.01) [c] |
| P3 | 24.5 (1.20) [e] | 46.3 (2.35) [cb] | 0.19 (0.01) [ed] | 3.5 (0.17) [c] | 0.30 (0.01) [fe] | 147.75 (7.35) [e] | 0.50 (0.02) [a] | 0.31 (0.01) [c] | 0.50 (0.02) [a] | 0.16 (0.01) [d] |
| P4 | 35.8 (1.72) [d] | 40.9 (1.99) [c] | 0.74 (0.03) [ed] | 2.99 (0.15) [dc] | 0.68 (0.03) [ed] | 66.89 (3.38) [f] | nd | 0.22 (0.01) [d] | nd | 0.26 (0.01) [b] |
| P5 | 35.2 (1.67) [d] | 27.4 (1.33) [d] | 0.14 (0.00) [e] | 1.70 (0.08) [e] | 1.52 (0.07) [c] | nd | nd | 0.16 (0.00) [e] | 0.41 (0.02) [cb] | 0.29 (0.02) [b] |
| P6 | 30.3 (1.59) [ed] | 46.1 (2.30) [cb] | nd | 1.84 (0.08) [e] | 0.09 (0.00) [f] | 242.38 (11.8) [c] | nd | 0.15 (0.00) [e] | 0.32 (0.01) [d] | 0.18 (0.01) [dc] |
| P7 | 36.9 (1.76) [d] | nd | 0.35 (0.02) [b] | 1.21 (0.06) [e] | 0.07 (0.00) [f] | 195.10 (9.97) [d] | nd | 0.13 (0.00) [e] | 0.36 (0.02) [dc] | 0.19 (0.01) [dc] |
| P8 | 66.9 (3.40) [b] | nd | 0.25 (0.01) [c] | 9.00 (0.46) [a] | 0.93 (0.05) [d] | nd | nd | 0.14 (0.01) [e] | 0.35 (0.02) [dc] | 0.26 (0.01) [b] |
| P9 | 32.2 (1.59) [d] | 41.3 (2.01) [b] | 0.56 (0.03) [a] | 2.74 (0.13) [d] | 0.59 (0.03) [ed] | 130.35 (6.76) [e] | nd | 0.16 (0.01) [e] | 0.36 (0.02) [dc] | 0.20 (0.01) [c] |
| P10 | 46.8 (2.22) [c] | nd | 0.15 (0.01) [e] | 9.05 (0.44) [a] | 1.51 (0.07) [c] | 60.73 (3.02) [f] | nd | 0.14 (0.00) [e] | 0.46 (0.02) [ba] | 0.29 (0.01) [b] |

The values are mean of three samples (n = 3, standard deviation in parentheses), analyzed individually in triplicate. Different letters within a column denote significant differences ($p < 0.05$); nd = not detectable.

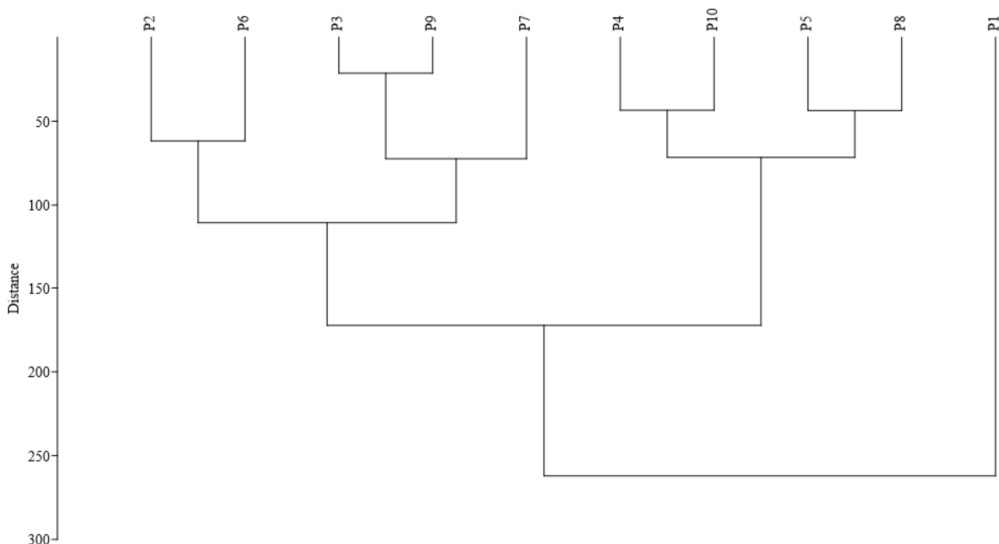

**Figure 5.** Hierarchical clustering of the honey samples based on physico-chemical parameters (Euclidean distance, *r* = 0.76).

Low honey pH and acidic substances, such as aromatic acids and royal jelly acids, have been reported to play an important role on antibacterial activity of honeys [5,46]. In the present study (Table 4), the pH value range between 3.78 (P1)–4.21 (P6), and total acidity value between 15.5 (P4)–65.2 (P8) meq/kg. The recommended value for free acidity, as stated in the European regulations, is 50 mEq/kg [47]. Higher amounts in free acidity were obtained in sample P8 (34.1 meq/kg), followed by P1 (25.3 meq/kg). Higher values are indication of incipient sugar fermentation, leading to the presence of acetic acid, formed by alcohol hydrolysis [48].

**Table 4.** pH and acidity results from raw honey samples.

| Honey Samples | pH | Free Acidity (meq/kg) | Lactone Acidity (meq/kg) | Total Acidity (meq/kg) |
|---|---|---|---|---|
| P1 | 3.78 (0.19) [a] | 25.3 (1.27) [b] | 23.3 (1.11) [b] | 48.6 (2.38) [b] |
| P2 | 4.17 (0.20) [a] | 13.5 (0.68) [d] | 10.1 (0.52) [f] | 23.6 (1.20) [ed] |
| P3 | 4.19 (0.21) [a] | 12.4 (0.61) [ed] | 14.1 (0.71) [ed] | 26.6 (1.28) [d] |
| P4 | 4.00 (0.20) [a] | 8.93 (0.44) [f] | 6.63 (0.34) [g] | 15.5 (0.77) [f] |
| P5 | 3.92 (0.20) [a] | 18.0 (0.92) [c] | 17.5 (0.90) [c] | 35.6 (1.74) [c] |
| P6 | 4.21 (0.20) [a] | 10.1 (0.52) [fe] | 9.71 (0.46) [f] | 19.8 (1.04) [fe] |
| P7 | 3.97 (0.20) [a] | 19.0 (0.92) [c] | 16.5 (0.79) [dc] | 35.6 (1.81) [c] |
| P8 | 3.84 (0.19) [a] | 34.1 (1.77) [a] | 31.6 (1.53) [a] | 65.2 (3.17) [a] |
| P9 | 3.9 (0.20) [a] | 20.1 (0.97) [c] | 14.2 (0.71) [ed] | 34.4 (1.66) [c] |
| P10 | 3.89 (0.18) [a] | 18.7 (0.91) [c] | 13.3 (0.67) [e] | 32.0 (1.54) [c] |

The values are mean of three samples (n = 3, standard deviation in parentheses), analyzed individually in triplicate. Different letters within a column denote significant differences ($p < 0.05$).

### 3.3. Quality Parameters of Raw Honey Samples

HMF is a parameter of honey freshness, since it is absent or present in trace amounts in fresh honeys. Furthermore, the HMF content in honey is established by the Codex Alimentarius Standard Commission [42] at 40 mg/kg. The amount of HMF identified in the studied raw honey samples ranged between $1.65 \pm 0.08$ mg/kg (P6) and $19.7 \pm 0.99$ mg/kg (P9).

Diastase is also considered an indicator of honey freshness for authentic samples or possible adulteration with inverted sugars, when the content is very low [49]. The values obtained in our study were between $41.37 \pm 2.06$ (P8) and $9.12 \pm 0.43$ (P9).

Production of honey without residues showed Good Apicultural Practice. Antibiotic residues can originate from treatments against the brood diseases American Foul Brood (AFB) or European Foul Brood (EFB). Treatments with antibiotics are not allowed in the EU, while in many other countries they are widely used [50]. The levels of oxytetracycline and tetracycline were studied for the raw honey samples. Only one sample (P5) presented 95 µg/kg oxytetracycline. All other honeys were free of antibiotics.

### 3.4. Microbiological Results of Raw Honey Samples

In five raw honey samples (P1, P4, P5, P9, P10), we have identified probiotic bacteria (Table 5), processed further by DNA extraction (Figure 6) and BLASTn analysis of Bacteria 16S sequence. The probiotic bacteria were: *Bacillus mycoides* (P1), *Bacillus thuringiensis* (P4), *Bacillus amyloliquefaciens* (P5), *Bacillus subtilis* and *Bacillus velezensis* (P9), and *B. thuringiensis* and *Bacillus velezensis* (P10).

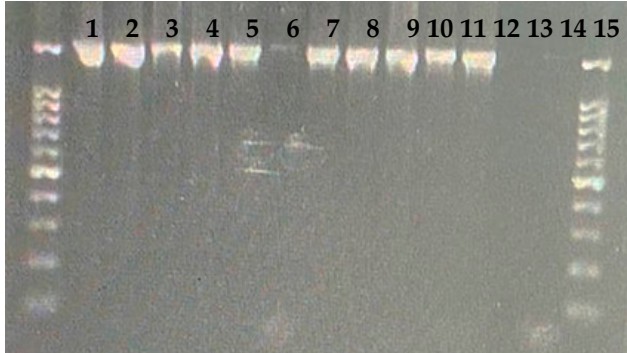

**Figure 6.** PCR amplification of bacteria DNA (1481 bp) Lines 1 and 15: 1500 bp DNA ladder, lines 2–6 and 8–12 bacterial DNA positive amplification (~1481 bp), lines 7 and 14 positive control, line 13 negative control.

**Table 5.** NTG quantification, probiotic, and other species identification in the raw honey samples analyzed.

| Raw Honey Samples | NTG CFU/mL | Identified Species |
|---|---|---|
| P1 | $1 \times 10^2$ | *Bacillus mycoides* |
| P2 | 0 | - |
| P3 | $100 \times 10^6$ | *Ewingella americana, Staphylococcus hominis* ssp. *hominis* |
| P4 | $51 \times 10^4$ | *Staphylococcus epidermidis, Bacillus thuringiensis, Aerococcus viridans* |
| P5 | $150 \times 10^4$ | *Bacillus amyloliquefaciens* * |
| P6 | $25 \times 10^4$ | *Staphylococcus hominis* ssp. *hominis, Candida glabrata* |
| P7 | 0 | - |
| P8 | $1 \times 10^2$ | *Leuconostoc mesenteroides* ssp. *cremoris* |
| P9 | $20 \times 10^2$ | *Bacillus subtilis, Bacillus velezensis* * |
| P10 | $28 \times 10^3$ | *Corynebacterium amycolatum, Staphylococcus haemolyticus, Staphylococcus warneri, B. thuringiensis *, Bacillus velezensis *, Candida glabrata* |

* Identified by BLASTn analysis of Bacteria 16S sequence.

In addition to probiotic bacteria, other species have been identified in the analyzed raw honey samples (Table 5). The NTG value was between $1 \times 10^2$ (P1, P8) and $100 \times 10^6$ (P3) CFU/mL. This aspect means that the beekeepers do not respect the Good Apicultural Practice and a correct hygiene in their apiary. No bacteria were identified in P2 and P7 samples.

### 3.5. Antibacterial Activity for the Identified Probiotic Bacteria

*Bacillus subtilis* complex species (*B. subtilis, B. amyloliquefaciens, B. velezensis*), identified in samples 1, 4, 5, 6, and 8, seems to have antibacterial activity especially against Gram-negative bacteria: *Pseudomonas aeruginosa, Escherichia coli*, followed by *Salmonella enteritidis*, but also against Gram-positive: *Listeria monocytogenes* and less against *Staphylococcus aureus*. No effect was observed against *Enterococcus faecalis*.

Antimicrobial activity of *Bacillus spp.* isolates against pathogenic bacteria is presented in Table 6 and Figure 7.

**Table 6.** Antimicrobial effect of Bacillus spp. isolates from the raw honey samples (zone of inhibition, mm).

| Antimicrobial Effect | 1 | 2 | 3 | 4 | 5 | 6 | 7 | 8 |
|---|---|---|---|---|---|---|---|---|
| *Enterococcus faecalis* | 0 | 0 | 0 | 0 | 0 | 0 | 0 | 0 |
| *Staphylococcus aureus* | 2.10 (0.11) [c] | 0 | 0 | 1.80 (0.09) [d] | 1.70 (0.38) [c] | 1.80 (0.08) [c] | 2.40 (0.12) [a] | 1.40 (0.07) [c] |
| *Listeria monocytogenes* | 3.4 (0.17) [b] | 1.8 (0.09) [a] | 1.9 (0.09) [a] | 3.50 (0.18) [b] | 2.60 (0.13) [b] | 2.80 (0.14) [b] | 2.20 (0.10) [b] | 1.80 (0.08) [c] |
| *Esherichia coli* | 3.4 (0.17) [b] | 0 | 0 | 3.40 (0.16) [cb] | 2.80 (0.13) [b] | 2.80 (0.15) [b] | 0 | 2.40 (0.12) [b] |
| *Salmonella enteritidis* | 3.2 (0.15) [b] | 0 | 0 | 2.99 (0.15) [c] | 2.80 (0.14) [b] | 2.80 (0.15) [b] | 0 | 2.60 (0.13) [b] |
| *Pseudomonas aeruginosa* | 5.2 (0.26) [a] | 0 | 0 | 4.90 (0.23) [a] | 4.50 (0.21) [a] | 4.40 (0.23) [a] | 0 | 6.40 (0.31) [a] |

The values are mean of three samples (n = 3, standard deviation in parentheses), analyzed individualy in triplicate. Different letters within a column denote significant differences ($p < 0.05$). 1 = B. subtilis, 2 = B. thuringiensis 1, 3 = B. mycoides, 4 = B. amyloliquefaciens, 5 = B. velezensis 1, 6 = B. velezensis 2, 7 = B. thuringiensis 2, 8 = B. velezensis 3, N = negative control.

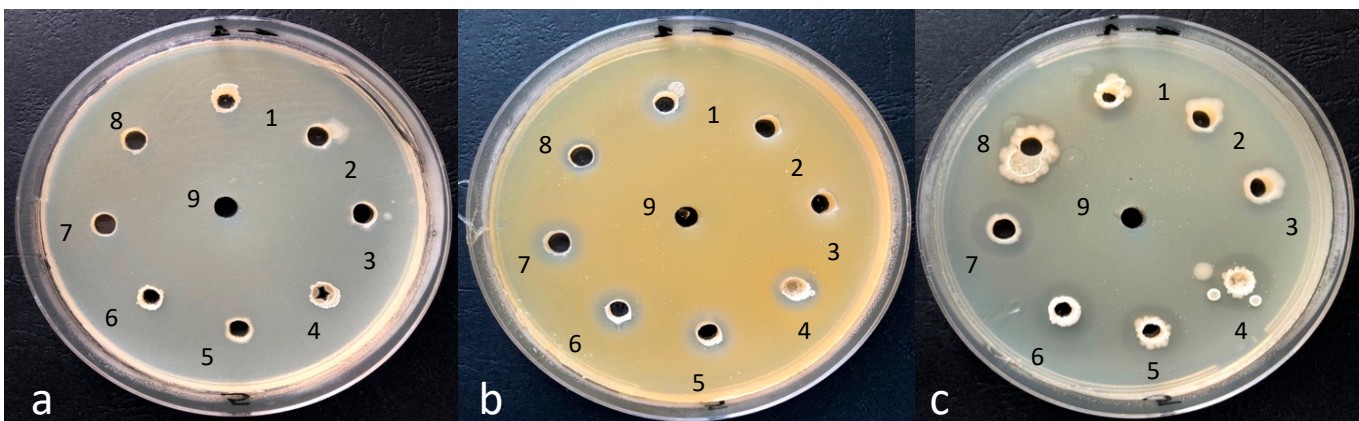

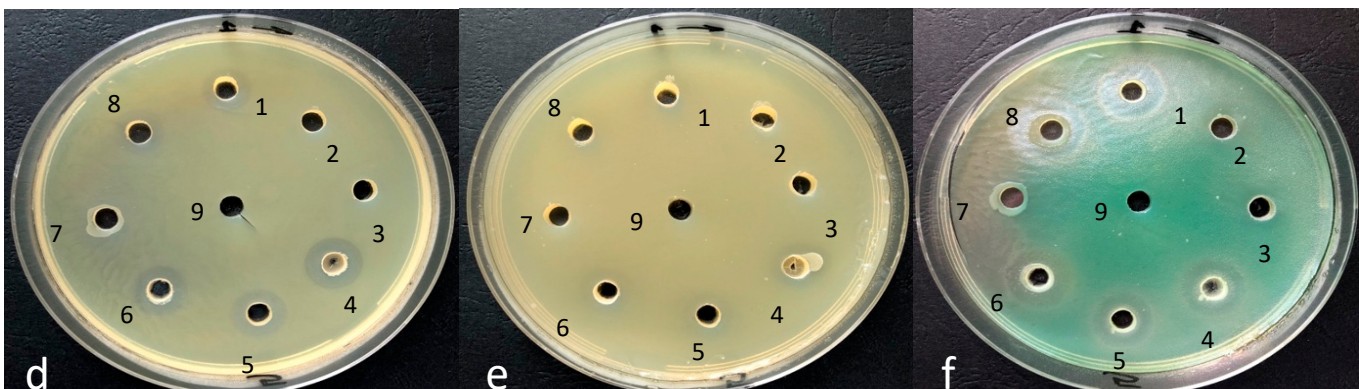

**Figure 7.** Antimicrobial activity of Bacillus spp. strains (1 = B. subtilis, 2 = B. thuringiensis 1, 3 = B. mycoides, 4 = B. amyloliquefaciens, 5 = B. velezensis 1, 6 = B. velezensis 2, 7 = B. thuringiensis 2, 8 = B. velezensis 3, N = negative control-central) against Enterococcus faecalis (**a**), Staphylococcus aureus (**b**), Listeria monocytogenes (**c**), Esherichia coli (**d**), Salmonella enteritidis (**e**), Pseudomonas aeruginosa (**f**).

## 4. Discussion

This study represents a screening of some Romanian raw honeys collected from different geographic area and investigated the chemical characterization, microbiological examination and probiotic effect for these honeys. The general effects of geographic origin on the chemical and microbiological characterization for honey have been reported previously [1,5–8,16,30,51–54]. From the ten different geographical locations, the botanical origin shows similarities, and, according to palynological examination and chemical characterization, the samples can be framed as polyfloral honey. Generally, this type of honey has water content below 20% [51,52], which can be observed also in this study. Regarding the sugar spectrum, the higher concentrations in fructose and glucose was registered for west (P9) and south-east (P8) of the country, while sucrose was in higher amounts in north-west of the country (P5 and P1). Fructooligosaccharides (sucrose), but also other oligosaccharides, can contribute as a nutritional ingredient, together with other prebiotic bacteria [9] from honeybee gut, to growing the probiotic bacteria, which have efficacy against pathogens [55]. Another important parameter for stimulating the growth and/or activity of the probiotic bacteria is the acidity. These bacteria need low acidity for development. The raw honey samples with low acidity and pH was registered in north-west (P1) and south-east (P8) of the country. The free acidity content of honey describes the presence and amount of organic acids, like butyric, acetic, formic, lactic, succinic, pyrogutamic, malic, and citric acids in equilibrium with their corresponding lactones, or internal esters, and some inorganic ions, such as phosphate [56,57]. Higher values were registered in samples from south-east (P8) of the country. The quality of honey is very important when this bee product is used as

natural antibiotic or as a nutritional supplement. In this context, we evaluated the HMF, diastase, contaminants from the class of antibiotics, and heavy metals from analyzed honey samples. In addition, in the north-west (P3) and in the center (P10) of the country, we found higher levels of Zn. This microelement together with the prebiotic and probiotic bacteria from honey increasing its antibacterial activity.

After the microbiological examination seven *Bacillus spp.* isolates were identified in the raw honey samples: in the north-west (P1, P5) of the country: *Bacillus mycoides* and *Bacillus amyloliquefaciens;* in the north (P4) of the country: *Bacillus thuringiensis*; in the west of the country (P9): *Bacillus subtilis, Bacillus velezensis*; and in the center (P10) of the country: *B. thuringiensis, Bacillus velezensis. Leuconostoc mesenteroides* ssp *cremoris* [58] was identified in a single sample. It is recognized as a potential probiotic, and its effect on *Candida albicans* [59] was studied before. Probiotic bacteria can be found in raw, unfiltered honey, including Lactobacilli and Bifidobacteria [17]. Esawy et al. (2012) [17] also found and isolated from different honey sources, probiotic bacteria, such as *B.subtilis*, *B.licheniforms*, *B. amyloliquefaciens*, *B. thuringiensis*, *B. cereus*, *B. pseudomycoides*, and *B. mycoides*. In the present study, no Lactobacilli or Bifidobacteria were identified. In addition, opportunistic pathogens were identified: *Ewingella americana* [60], *Staphylococcus hominis* ssp. *Hominis* [61], *S. haemolyticus*, *S. warneri*, *S. epidermidis* [62], *Aerococcus viridans* [63], *Leuconostoc mesenteroides* ssp *cremoris* [58,64], *Corynebacterium amycolatum* [65], and *Candida glabrata* [66]. Despite the pathogenic potential of these species, the low value of NTG in the tested samples indicate a low risk for consumers, with the exception of the sample P3, in which two potentially pathogenic bacteria in a high concentration ($>10^6$ CFU/mL) were detected.

*Bacillus subtilis* complex species identified have antibacterial activity against Gram-positive and Gram-negative bacteria, which we have demonstrated in this research study. In addition, literature studied [67] showed antimicrobial activity of probiotic bacteria (*B. endophyticus* and *B. subtilis*) against different pathogenic species. *B. subtilis* had higher antimicrobial activity against all tested bacteria, except *Staphylococcus aureus* and *C. albicans*, while *B. endophyticus* showed antimicrobial activities against *B.cereus*, *C. albicans*, *C. trobicales*, and *Saccharomyces cerevisiae*. In addition, *B. subtilis* had antimicrobial activity against *Staphylococcus aureus*. Sabate et al. (2012) [68] demonstrated that in vivo administration of *B. subtilis sub*sp. *subtilis* Mori2 had three beneficial effects on bee colonies: an increase in open and operculated brood, greater accumulation of honey compared to the control hives, and a healthier hive, due to the reduction of *Varroa* and *Nosema* incident rates.

## 5. Conclusions

In the samples from the north-west of the country (P1 and P5), being distinct from the others due to its high content in minerals, a high content in sucrose and lower in lipids and trehalose is noticed. More studies on different maturity stages of honey are needed to certify the level of different constituents and their connection to health promoting properties related to the presence of probiotics bacteria.

Honey possesses natural antibacterial activity due to factors, such as botanical and geographical origin, sugar content, acidity, and other chemical compounds. In this study, we demonstrated that raw honey from different area could have probiotic potential, predominantly from the *Bacillus subtilis* complex species. In conclusion, Romanian raw honey can be a potential reservoir of probiotics, which confer a health benefit for consumers.

**Author Contributions:** All authors contribute equally at the manuscript preparation. All authors have read and agreed to the published version of the manuscript.

**Funding:** This work was supported by a grant of the Ministry of Research, Innovation and Digitization, CNCS/CCCDI—UEFISCDI, project number PN-III-P1-1.1-PD-2019-0498, within PNCDI III.

**Institutional Review Board Statement:** Not applicable.

**Informed Consent Statement:** Not applicable.

**Data Availability Statement:** Not applicable.

**Conflicts of Interest:** The authors declare no conflict of interest.

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
