# Peer review of "Screening of Some Romanian Raw Honeys and Their Probiotic Potential Evaluation"

_applsci, doi:10.3390/app11135816_

Round 1

Reviewer 1 Report

In their manuscript Pasca et al. describe the molecular and microbial composition of raw honey from different sources in Romania. This is a well performed study. Only the English style should be improved and at the end of the introduction it should be specified in more detail why the probiotic strains might be carrying the risk of transferring antibiotic resistance.

Author Response

Reviewer 1:

In their manuscript Pasca et al. describe the molecular and microbial composition of raw honey from different sources in Romania. This is a well performed study. Only the English style should be improved and at the end of the introduction it should be specified in more detail why the probiotic strains might be carrying the risk of transferring antibiotic resistance.

Response to Reviewer 1:

We thank the reviewer for its pertinent observations that made our manuscript improved. The punctual answer is:

  1. We improved the English style for the manuscript.
  2. Generally, all honey samples meet the standard values for chemical composition. Although, one sample having 7.44 % sucrose, was found to have also probiotics bacteria from the genus Bacillus, because sucrose is a substrate for probiotics development.
  3. The probiotics don`t carry the risk for antibiotics resistance, they are used exactly for the opposite: the organism shall not develop resistance towards antibiotics.

Reviewer 2 Report

Extensive research has been done, and the manuscript is well written. In the paper, the characterization of different honey is described and compared.

The paper can be published after minor revision.

Comments:

  • In the Materials and Methods section, the procedure of the determination of Total lipids should not only be referenced but also described in detail.
  • Figure 4 is not explained anyhow. Please correct it.
  • In figure 3, only the peak of sucrose is indicated, however, Table 2 indicates more sugars. Please, show a complete chromatogram with the peaks.
  • In figure 2, a dendrogram is generated according to the Melisopalinological analysis. What would be extremely useful is the generation and description of another dendrogram using physicochemical properties only.

Author Response

Reviewer 2:

Extensive research has been done, and the manuscript is well written. In the paper, the characterization of different honey is described and compared.

The paper can be published after minor revision.

Comments:

  • In the Materials and Methods section, the procedure of the determination of Total lipids should not only be referenced but also described in detail.
  • Figure 4 is not explained anyhow. Please correct it.
  • In figure 3, only the peak of sucrose is indicated, however, Table 2 indicates more sugars. Please, show a complete chromatogram with the peaks.
  • In figure 2, a dendrogram is generated according to the Melisopalinological analysis. What would be extremely useful is the generation and description of another dendrogram using physicochemical properties only.

Response to Reviewer 2:

We thank the reviewer for its pertinent observations that made our manuscript improved. The punctual answer is:

  1. Briefly, 2 g of honey samples were weighted on filter paper and also the extraction glasses that containing 2 boiling stones, will be weighted and after that together with the cartridge and the solvent (90 ml etil-eteric) will be fixed in PTFE cylinders. The method is set from the multistat: extraction temperature 140 °C, extraction time 5h, washing 30 min, solvent evaporation in hot air flow 10 min. After extraction ends, extraction glasses will be placed in the oven at 60 °C, for one half hour, to eliminate traces of solvent and after cooling will be weighted and the result were expressed as percent.
  2. The legend for figure 4: Lines 1 and 15: 1500 bp DNA ladder, lines 2-6 and 8-12  bacterial DNA positive amplification (~1481bp), lines 7 and 14 positive control, line 13 negative control.

3. Chromatogram of the identified carbohydrates in the raw honey samples was made.

4. As shown in Figure 5, the dendrogram obtained from the Cluster analysis (Euclidean distance) reveals the formation of two clusters with the exception of sample P1 which is an ‘outlier’, being distinct from the others due to its high content in the minerals K, Ca and Mg. At the same time, a high content in sucrose and lower in lipids and trehalose is noticed. The following cluster comprises the samples P8 and P5, followed by samples P10 and P4 which display similar concentrations in fructose, glucose and lipids, as well as in Ca, Se and Cr. The next sub-cluster comprises samples P7, P9 and P3 which have similar protein values, as well as in the mineral K. The last sub-cluster comprises samples P6 and P2 with relatively close values in fructose, glucose and moisture content. The same is noticed in the minerals Ca, K, but also in those with lower values such as Zn and Cr.

Reviewer 3 Report

The manuscript "Screening of Some Romanian Raw Honeys and their Probiotic Potential Evaluation" is interesting and very well written. The test methods used are described in great detail. Correct statistical analyzes are used. Honey has been characterized very carefully. The discussion of the results is very insightful.

Detailed comments:

More research results can be given in the abstract.

Figure 3 - Can a legend be added which color represents which sample?

Table 3 - explain under the table "nd"

Table 5 - NTG CFU / mL results should be reported in exponential notation; the same in the description of these results. What was the detection limit of microorganisms in the samples?

The chapter Conclusion should be redrafted and supplemented with the results obtained. 

Author Response

Reviewer 3:

The manuscript "Screening of Some Romanian Raw Honeys and their Probiotic Potential Evaluation" is interesting and very well written. The test methods used are described in great detail. Correct statistical analyzes are used. Honey has been characterized very carefully. The discussion of the results is very insightful.

Detailed comments:

More research results can be given in the abstract.

Figure 3 - Can a legend be added which color represents which sample?

Table 3 - explain under the table "nd"

Table 5 - NTG CFU / mL results should be reported in exponential notation; the same in the description of these results. What was the detection limit of microorganisms in the samples?

The chapter Conclusion should be redrafted and supplemented with the results obtained.

Response to Reviewer 3:

We thank the reviewer for its pertinent observations that made our manuscript improved. The punctual answer is:

  1. I put the legend for this chromatogram.
  2. Nd means not detectable, I explain in the manuscript
  3. NTG CFU/ML are expressed as exponential numbers in the manuscript. I express now with superscript. The detection limit of microorganisms in the samples is 1 CFU/ml.
  4. The samples from north-west of the country (P1 and P5), being distinct from the others due to its high content in minerals, a high content in sucrose and lower in lipids and trehalose is noticed

Reviewer 4 Report

The article is interesting and valuable. Characterization of raw honeys from different geographical origins in Romania associated with analysis of chemical composition, microbiological examination and probiotic potential evaluation provided valuable information. Therefore, the undertaken research topic by Authors seems to be very interesting.

The reviewer suggests minor revisions. The list of suggestions and remarks are below:

  1. The cited information listed (Introduction section; lines 47-54) was not supported by any literature items.
    Literature should be provided.

    2.The Authors of the article did not provide a convincing justification for the choice of honey as the leading issue of the article.

    3. The authors of the text use the following expression:

    High sucrose values ​​in honeys are related to its botanical origin, honey maturity,
    high nectar flux or artificial feeding of bees.

    Authors should explain the meaning of the phrase: "High sucrose values"

    As well as the mentioned relation: "High sucrose values ​​in honeys are related to its botanical origin, honey maturity,
    high nectar flux or artificial feeding of bees" - remains unclear - it needs at least short explanation.

    4. The Authors of the manuscript did not provide the information concerning the standardisation of the "Microbiological examination of raw honey samples" procedure

    5. The description of Figure 5 should be completed by assigning specific samples to the colour of the curves presented in the chromatogram

Author Response

Reviewer 4:

The article is interesting and valuable. Characterization of raw honeys from different geographical origins in Romania associated with analysis of chemical composition, microbiological examination and probiotic potential evaluation provided valuable information. Therefore, the undertaken research topic by Authors seems to be very interesting.

The reviewer suggests minor revisions. The list of suggestions and remarks are below:

  1. The cited information listed (Introduction section; lines 47-54) was not supported by any literature items.
    Literature should be provided.

    2.The Authors of the article did not provide a convincing justification for the choice of honey as the leading issue of the article.

    3. The authors of the text use the following expression:

    High sucrose values ​​in honeys are related to its botanical origin, honey maturity,
    high nectar flux or artificial feeding of bees.

    Authors should explain the meaning of the phrase: "High sucrose values"

    As well as the mentioned relation: "High sucrose values ​​in honeys are related to its botanical origin, honey maturity,
    high nectar flux or artificial feeding of bees" - remains unclear - it needs at least short explanation.

    4. The Authors of the manuscript did not provide the information concerning the standardisation of the "Microbiological examination of raw honey samples" procedure

    5. The description of Figure 5 should be completed by assigning specific samples to the colour of the curves presented in the chromatogram

Response to Reviewer 4:

We thank the reviewer for its pertinent observations that made our manuscript improved. The punctual answer is:

  1. The lines 47-54 represents the definition of honey by European Union Council Directive 2001/110/EC, and we have there the reference
  2. In this research we choose to study honey because it is the most consumed bee product.
  3. Sucrose is a fructooligosaccharide, which is prebiotics, and prebiotics are substrates for develop probiotics. If the concentration in fructooligosaccharides are higher the probiotics are developing fast. High sucrose values in honeys are related to its botanical origin, for exemple the honey dew or different honey maturity stages, or high nectar flux or artificial feeding of honey bees with sugar syrup.
  4. Microbiological examination of honey was performed following the methods described by Hamdy et al., 2017, adapted method in our laboratory.
  5. I put the legend for the cromatogram

Reviewer 5 Report

Dear Authors, It is a very well written comprehensive medicinal properties of honey. Honey is the first known sweetener for the mankind. It has significant biomedicinal potential. One thing which I noticed is what chemical composition is required for the biomedicinal value may be discussed and the structures of the chemical compositions may help the mediicinal chemists to connect efficiently. But , it is very good comprehensive study.

Author Response

Reviewer 5:

Dear Authors, It is a very well written comprehensive medicinal properties of honey. Honey is the first known sweetener for the mankind. It has significant biomedicinal potential. One thing which I noticed is what chemical composition is required for the biomedicinal value may be discussed and the structures of the chemical compositions may help the mediicinal chemists to connect efficiently. But , it is very good comprehensive study.

Response to Reviewer 5:

We thank the reviewer for its pertinent observations that made our manuscript improved. The punctual answer is:

Our main goal of the study was not to analyze honey from a medicinal point of view. Honey comprises mainly from sugars, the rest of the components are present in very small amounts and all components have their own importance for the medicinal value of honey. All honey was tested for the presence of antibiotics, because for medicinal purposes, honey must be free of contaminants (page 17 line 467-468).